# Learning Sparse Neural Networks
# via Sensitivity-Driven Regularization

**Enzo Tartaglione**
Politecnico di Torino
Torino, Italy
tartaglioneenzo@gmail.com

**Skjalg Lepsøy**
Nuance Communications
Torino, Italy

**Attilio Fiandrotti**
Politecnico di Torino, Torino, Italy
Télécom ParisTech, Paris, France

**Gianluca Francini**
Telecom Italia
Torino, Italy

## Abstract

The ever-increasing number of parameters in deep neural networks poses challenges for memory-limited applications. Regularize-and-prune methods aim at meeting these challenges by sparsifying the network weights. In this context we quantify the output *sensitivity* to the parameters (i.e. their relevance to the network output) and introduce a regularization term that gradually lowers the absolute value of parameters with low sensitivity. Thus, a very large fraction of the parameters approach zero and are eventually set to zero by simple thresholding. Our method surpasses most of the recent techniques both in terms of sparsity and error rates. In some cases, the method reaches twice the sparsity obtained by other techniques at equal error rates.

## 1 Introduction

Deep neural networks achieve state-of-the-art performance in a number of tasks by means of *complex* architectures. Let us define the complexity of a neural network as the number of its learnable parameters. The complexity of architectures such as VGGNet [1] and the SENet-154 [2] lies in the order of $10^8$ parameters, hindering their deployability on portable and embedded devices, where storage, memory and bandwidth resources are limited.

The complexity of a neural network can be reduced by promoting *sparse* interconnection structures. Empirical evidence shows that deep architectures often require to be *over-parametrized* (having more parameters than training examples) in order to be successfully trained [3, 4, 5]. However, once input-output relations are properly represented by a complex network, such a network may form a starting point in order to find a simpler, sparser, but sufficient architecture [4, 5].

Recently, *regularization* has been proposed as a principle for promoting sparsity during training. In general, regularization replaces unstable (ill-posed) problems with nearby and stable (well-posed) ones by introducing additional information about what a solution should be like [6]. This is often done by adding a term $R$ to the original objective function $L$. Letting $\theta$ denote the network parameters and $\lambda$ the regularization factor, the problem

$$\text{minimize } L(\theta) \text{ with respect to } \theta \tag{1}$$

is recast as

$$\text{minimize } L(\theta) + \lambda R(\theta) \text{ with respect to } \theta. \tag{2}$$

Stability and generalization are strongly related or even equivalent, as shown by Mukherjee *et al.* [7]. Regularization therefore also helps ensure that a properly trained network generalizes well on unseen data.

Several known methods aim at reaching sparsity via regularization terms that are more or less specifically designed for the goal. Examples are found in [8, 9, 10].

The original contribution of this work is a regularization and pruning method that takes advantage of output *sensitivity* to each parameter. This measure quantifies the change in network output that is brought about by a change in the parameter. The proposed method gradually moves the less sensitive parameters towards zero, avoiding harmful modifications to sensitive, therefore important, parameters. When a parameter value approaches zero and drops below a threshold, the parameter is set to zero, yielding the desired sparsity of interconnections.

Furthermore, our method implies minimal computational overhead, since the sensitivity is a simple function of a by-product of back-propagation. Image classification experiments show that our method improves sparsity with respect to competing state-of-the-art techniques. According to our evidence, the method also improves generalization.

The rest of this paper is organized as follows. In Sec. 2 we review the relevant literature concerning sparse neural architectures. Next, in Sec. 3 we describe our supervised method for training a neural network such that its interconnection matrix is sparse. Then, in Sec. 4 we experiment with our proposed training scheme over different network architectures. The experiments show that our proposed method achieves a tenfold reduction in the network complexity while leaving the network performance unaffected. Finally, Sec. 5 draws the conclusions while providing further directions for future research.

## 2   Related work

Sparse neural architectures have been the focus of intense research recently due the advantages they entail. For example, Zhu *et al.* [11], have shown that a large sparse architecture improves the network generalization ability in a number of different scenarios. A number of approaches towards sparse interconnection matrices have been proposed. For example, Liu *et al.* [12] propose to recast multi-dimensional convolutional operations into bidimensional equivalents, resulting in a final reduction of the required parameters. Another approach involves the design of an object function to minimize the number of features in the convolutional layers. Wen *et al.* [8] propose a regularizer based on group lasso whose task is to cluster filters. However, such approaches are specific for convolutional layers, whereas the bulk of network complexity often lies in the fully connected layers.

A direct strategy to introduce sparsity in neural networks is $l_0$ regularization, which entails however solving a highly complex optimization problem (e.g., Louizos *et al.* [13]).

Recently, a technique based on soft weight sharing has been proposed to reduce the memory footprint of whole networks (Ullrich *et al.* [10]). However, it limits the number of the possible parameters values, resulting in sub-optimal network performance.

Another approach involves making input signals sparse in order to use smaller architectures. Inserting autoencoder layers at the begin of the neural network (Ranzato *et al.* [14]) or modeling of 'receptive fields' to preprocess input signals for image classification (Culurciello *et al.* [15]) are two clear examples of how a sparse, properly-correlated input signal can make the learning problem easier.

In the last few years, dropout techniques have also been employed to ease sparsification. Molchanov *et al.* [16] propose variational dropout to promote sparsity. This approach also provides a bayesian interpretation of gaussian dropout. A similar but differently principled approach was proposed by Theis *et al.* [17]. However, such a technique does not achieve in fully-connected architectures state-of-the-art test error.

The proposal of Han *et al.* [9] consists of steps that are similar to those of our method. It is a three-staged approach in which first, a network learns a coarse measurement of each connection importance, minimizing some target loss function; second, all the connections less important than a threshold are pruned; third and finally, the resulting network is retrained with standard backpropagation to learn the actual weights. An application of such a technique can be found in [18]. Their experiments

show reduced complexity for partially better performance achieved by avoiding network over-parametrization.

In this work, we propose to selectively prune each network parameter using the knowledge of sensitivity. Engelbrecht *et al.* [19] and Mrazova *et al.* [20, 21] previously proposed sensitivity-based strategies for learning sparse architectures. In their work, the sensitivity is however defined as the variation of the network output with respect to a variation of the network inputs. Conversely, in our work we define the sensitivity of a parameter as the variation of the network output with respect to the parameter, pruning parameters with low sensitivity as they contribute little to the network output.

# 3 Sensitivity-based regularization

In this section, we first formulate the sensitivity of a network with respect to a single network parameter. Next, we insert a sensitivity-based term in the update rule. Then, we derive a per-parameter general formulation of a regularization term based on sensitivity, having as particular case ReLU-activated neural networks. Finally, we propose a general training procedure aiming for sparsity. As we will experimentally see in Sec. 4, our technique not only sparsifies the network, but improves its generalization ability as well.

## 3.1 Some notation

Here we introduce the terminology and the notation used in the rest of this work. Let a feed-forward, acyclic, multi-layer artificial neural network be composed of $N$ layers, with $\mathbf{x}_{n-1}$ being the input of the $n$-th network layer and $\mathbf{x}_n$ its output, $n \in [1, N]$ integer. We identify with $n{=}0$ the *input layer*, $n = N$ the *output layer*, and other $n$ values indicate the *hidden layers*. The $n$-th layer has learnable parameters, indicated by $\mathbf{w}_n$ (which can be biases or weights).[1] In order to identify the $i$-th parameter at layer $n$, we write $w_{n,i}$.

The output of the $n$-th layer can be described as

$$\mathbf{x}_n = f_n\left[g_n\left(\mathbf{x}_{n-1}, \mathbf{w}_n\right)\right],\tag{3}$$

where $g_n(\cdot)$ is usually some affine function and $f_n(\cdot)$ is the *activation function* at layer $n$. In the following, $\mathbf{x}_0$ indicates the network input. Let us indicate the output of the network as $\mathbf{y} = \mathbf{x}_N \in \mathbb{R}^C$, with $C \in \mathbb{N}$. Similarly, $\mathbf{y}^\star$ indicates the target (expected) network output associated to $\mathbf{x}_0$.

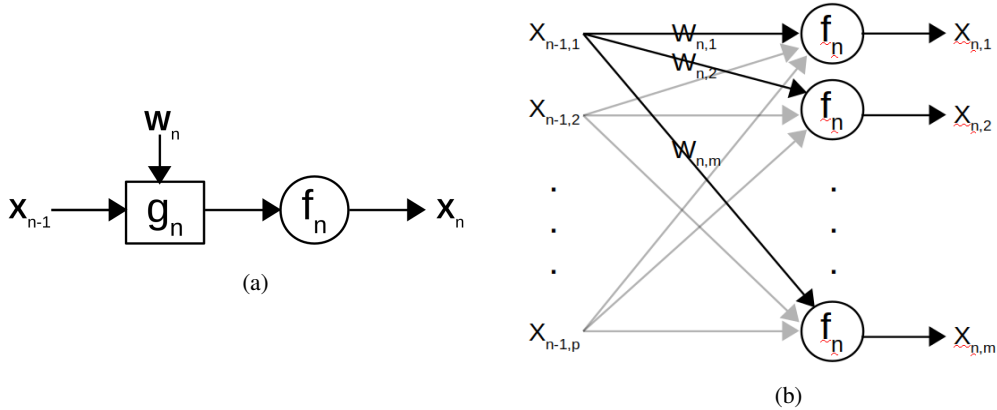

(a)

(b)

Figure 1: Generic layer representation (Fig. 1a) and the case of a fully connected layer in detail (Fig. 1b, here we have $\mathbf{w}_n \in \mathbb{R}^{m \times p}$). Biases may also be included.

## 3.2 Sensitivity definition

We are interested in evaluating how influential a generic parameter $w_{n,i}$ is to determine the $k$-th output of the network (given an input of the network).

Let the weight $w_{n,i}$ vary by a small amount $\Delta w_{n,i}$, such that the output varies by $\Delta \mathbf{y}$. For small $\Delta w_{n,i}$, we have, for each element,

$$\Delta y_k \approx \Delta w_{n,i} \frac{\partial y_k}{\partial w_{n,i}} \tag{4}$$

by a Taylor series expansion. A weighted sum of the variation in all output elements is then

$$\sum_{k=1}^{C} \alpha_k |\Delta y_k| = |\Delta w_{n,i}| \sum_{k=1}^{C} \alpha_k \left| \frac{\partial y_k}{\partial w_{n,i}} \right| \tag{5}$$

where $\alpha_k > 0$. The sum on the right-hand side is a key quantity for the regularization, so we define it as the *sensitivity*:

**Definition 1 (Sensitivity)** *The sensitivity of the network output with respect to the $(n,i)$-th network parameter is*

$$S(\mathbf{y}, w_{n,i}) = \sum_{k=1}^{C} \alpha_k \left| \frac{\partial y_k}{\partial w_{n,i}} \right|, \tag{6}$$

*where the coefficients $\alpha_k$ are positive and constant.*

The choice of coefficients $\alpha_k$ will depend on the application at hand. In Subsec. 3.5 we propose two choices of coefficients that will be used in the experiments.

If the sensitivity with respect to a given parameter is small, then a small change of that parameter towards zero causes a very small change in the network output. After such a change, and if the sensitivity computed at the new value still is small, then the parameter may be moved towards zero again. Such an operation can be paired naturally with a procedure like gradient descent, as we propose below. Towards this end, we introduce the *insensitivity* function $\bar{S}$

$$\bar{S}(\mathbf{y}, w_{n,i}) = 1 - S(\mathbf{y}, w_{n,i}) \tag{7}$$

The range of such a function is $(-\infty; 1]$ and the lower it is the more the parameter is relevant. We observe that having $\bar{S} < 0 \Leftrightarrow S > 1$ means that a weight change brings about an output change that is bigger than the weight change itself (5). In this case we say the output is *super-sensitive* to the weight. In our framework we are not interested in promoting the sparsity for such a class of parameters; on the contrary, they are very relevant for generating the output. We want to focus our attention towards all those parameters whose variation is not significantly felt by the output ($\sum_k \alpha_k |\Delta y_k| < \Delta w$), for which the output is *sub-sensitive* to them. Hence, we introduce a bounded insensitivity

$$\bar{S}_b(\mathbf{y}, w_{n,i}) = \max \left[ 0, \bar{S}(\mathbf{y}, w_{n,i}) \right] \tag{8}$$

having $\bar{S}_b \in [0, 1]$.

## 3.3 The update rule

As already hinted at, a parameter with small sensitivity may safely be moved some way towards zero. This can be done by subtracting a product of the parameter itself and its insensitivity measure (recall that $\bar{S}_b$ is between 0 and 1), appropriately scaled by some small factor $\lambda$.

Such a subtraction may be carried out simultaneously with the step towards steepest descent, effectively modifying SGD to incorporate the push of less 'important' parameters towards small values.

This brings us to the operation at the core of our method – the rule for updating each weight. At the $t$-th update iteration, the $i$-th weight in the $n$-th layer will be updated as

$$w_{n,i}^t := w_{n,i}^{t-1} - \eta \frac{\partial L}{\partial w_{n,i}^{t-1}} - \lambda w_{n,i}^{t-1} \bar{S}_b(\mathbf{y}, w_{n,i}^{t-1}) \qquad (9)$$

where $L$ is a loss function, as in (1) and (2). Here we see why the bounded insensitivity is not allowed to become negative: this would allow to push some weights (the super-sensitive ones) away from zero.

Below we show that each of the two correction terms dominates over the other in different phases of the training. The supplementary material treats this matter in more detail.

The derivative of the first correction term in (9) wrt. to the weight (disregarding $\eta$) can be factorized as

$$\frac{\partial L}{\partial w_{n,i}} = \frac{\partial L}{\partial \mathbf{y}} \frac{\partial \mathbf{y}}{\partial w_{n,i}} \qquad (10)$$

which is a scalar product of two vectors: the derivative of the loss with respect to the output elements and the derivative of the output elements with respect to the parameter in question. By the Hölder inequality, we have that

$$\left| \frac{\partial L}{\partial w_{n,i}} \right| \le \max_k \left| \frac{\partial L}{\partial y_k} \right| \left\| \frac{\partial \mathbf{y}}{\partial w_{n,i}} \right\|_1 . \qquad (11)$$

Furthermore, if the loss function $L$ is the composition of the cross-entropy and the softmax function, the derivative of $L$ with respect to any $y_k$ cannot exceed 1 in absolute value. The inequality in eq.11 then simplifies to

$$\left| \frac{\partial L}{\partial w_{n,i}} \right| \le \left\| \frac{\partial \mathbf{y}}{\partial w_{n,i}} \right\|_1 . \qquad (12)$$

We note that the $l_1$ norm on the right is proportional to the sensitivity of (6), provided that all coefficients $\alpha_k$ are equal (as in (17) in a later section). Otherwise the $l_1$ norm is *equivalent* to the sensitivity. For the following, we think of the $l_1$ norm on the right in eq.12 as a multiple of the sensitivity.

By (7), the *insensitivity* is complementary to the sensitivity. The *bounded insensitivity* is simply a restriction of the insensitivity to non-negative values (8).

Now we return to the two correction terms in the update rule of (9). If the first correction term is large, then by (12) also the sensitivity must be large. A large sensitivity implies a small (or zero) bounded insensitivity. Therefore a large first correction term implies a small or zero second correction term. This typically happens in early phases of training, when the loss can be greatly reduced by changing a weight, i.e. when $\frac{\partial L}{\partial w_{n,i}}$ is large.

Conversely, if the loss function is near a minimum, then the first correction term is very small. In this situation, the above equations do not imply anything about the magnitude of the sensitivity. The bounded insensitivity may be near 1 for some weights, thus the second correction term will dominate. These weights will be moved towards zero in proportion to $\lambda$. Sec. 4 shows that this indeed happens for a large number of weights.

The parameter cannot be moved all the way to zero in one update, as the insensitivity may change when $w_{n,i}$ changes; it must be recomputed at each new updated value of the parameter. The factor $\lambda$ should therefore be (much) smaller than 1.

### 3.4 Cost function formulation

The update rule of (9) does provide the "additional information" typical of regularization methods. Indeed, this method amounts to the addition of a regularization function $R$ to an original loss function, as in (1). Since (9) specifies how a parameter is updated through the *derivative* of $R$, an integration of the update term will 'restore' the regularization term. The result is readily interpreted for ReLU-activated networks [3].

Towards this end, we define the overall regularization term as a sum over all parameters

$$R(\theta) = \sum_i \sum_n R_{n,i}(w_{n,i}) \tag{13}$$

and integrate each term over $w_{n,i}$

$$R_{n,i}(w_{n,i}) = \int w_{n,i} \bar{S}_b(\mathbf{y}, w_{n,i}) dw_{n,i}. \tag{14}$$

If we solve (14) we find

$$R_{n,i}(w_{n,i}) = H\left[\bar{S}(\mathbf{y}, w_{n,i})\right] \frac{w_{n,i}^2}{2} \cdot \left[1 - \sum_{k=1}^{C} \alpha_k \mathrm{sign}\left(\frac{\partial y_k}{\partial w_{n,i}}\right) \sum_{m=1}^{\infty} -1^{m+1} \frac{w^{m-1}}{(m+1)!} \frac{\partial^m y_k}{\partial w_{n,i}^m}\right] \tag{15}$$

where $H(\cdot)$ is the Heaviside (one-step) function. (15) holds for any feedforward neural network having any activation function.

Now suppose that all activation functions are rectified linear units. Its derivative is the step function; the higher order derivatives are therefore zero. This results in dropping all the $m > 1$ terms in (15). Thus, the regularization term for ReLU-activated networks reduces to

$$R_{n,i}(w_{n,i}) = \frac{w_{n,i}^2}{2} \bar{S}(\mathbf{y}, w_{n,i}) \tag{16}$$

The first factor in this expression is the square of the weight, showing the relation to Tikhonov regularization. The other factor is a selection and damping mechanism. Only the sub-sensitive weights are influenced by the regularization – in proportion to their insensitivity.

## 3.5 Types of sensitivity

In general, (6) allows for different kinds of sensitivity, depending on the value assumed by $\alpha_k$. This freedom permits some adaptation to the learning problem in question.

If all the $k$ outputs assume the same "relevance" (all $\alpha_k = \frac{1}{C}$) we say we have an *unspecific* formulation

$$S^{unspec}(\mathbf{y}, w_{n,i}) = \frac{1}{C} \sum_{k=1}^{C} \left|\frac{\partial y_k}{\partial w_{n,i}}\right| \tag{17}$$

This formulation does not require any information about the training examples.

Another possibility, applicable to classification problems, does take into account some extra information. In this formulation we let only one term count, namely the one that corresponds to the desired output class for the given input $\mathbf{x}_0$. The factors $\alpha_k$ are therefore taken as the elements in the one-hot encoding for the desired output $\mathbf{y}^*$. In this case we speak of *specific* sensitivity:

$$S^{spec}(\mathbf{y}, \mathbf{y}^*, w_{n,i}) = \sum_{k=1}^{C} y_k^* \left|\frac{\partial y_k}{\partial w_{n,i}}\right| \tag{18}$$

The experiments in Sec. 4 regard classification problems, and we apply both of the above types of sensitivity.

## 3.6 Training procedure

Our technique ideally aims to put to zero a great number of parameters. However, according to our update rule (9), less sensitive parameters approach zero but seldom reach it exactly. For this reason, we introduce a threshold $T$. If

$$|w_{n,i}| < T$$

the method will *prune* it. According to this, the threshold in the very early stages must be kept to very low values (or must be introduced afterwards). Our training procedure is divided into two different steps:

Table 1: LeNet300 network trained over the MNIST dataset

| | Remaining parameters | | | | Memory | $\frac{|\theta|}{|\theta_{\neq 0}|}$ | Top-1 |
| | FC1 | FC2 | FC3 | Total | footprint | | error |
|---|---|---|---|---|---|---|---|
| Han *et al.* [9] | 8% | **9%** | **26%** | 21.76k | 87.04kB | 12.2x | 1.6% |
| Proposed ($S^{unspec}$) | **2.25%** | 11.93% | 69.3% | **9.55k** | **34.2kB** | **27.87x** | 1.65% |
| Proposed ($S^{spec}$) | 4.78% | 24.75% | 73.8% | 19.39k | 77.56kB | 13.73x | **1.56%** |
| Louizos *et al.* [13] | 9.95% | 9.68% | 33% | 26.64k | 106.57kB | 12.2x | 1.8% |
| SWS[10] | N/A | N/A | N/A | 11.19k | 44.76kB | 23x | 1.94% |
| Sparse VD[16] | 1.1% | 2.7% | 38% | 3.71k | 14.84kB | 68x | 1.92% |
| DNS[24] | 1.8% | 1.8% | **5.5%** | 4.72k | 18.88kB | 56x | 1.99% |
| Proposed ($S^{unspec}$) | **0.93%** | **1.12%** | 5.9% | **2.53k** | **10.12kB** | **103x** | 1.95% |
| Proposed ($S^{spec}$) | 1.12% | 1.88% | 13.4% | 3.26k | 13.06kB | 80x | 1.96% |

1. *Reaching the performance*: in this phase we train the network in order to get to the target performance. Here, any training procedure may be adopted: this makes our method suitable also for pre-trained models and, unlike other state-of-the-art techniques, can be applied at any moment of training.

2. *Sparsify*: thresholding is introduced and applied to the network. The learning process still advances but in the end of every training epoch all the weights of the network are thresholded. The procedure is stopped when the network performance drops below a given target performance.

## 4    Results

In this section we experiment with our regularization method in different supervised image classification tasks. Namely, we experiment training a number of well-known neural network architectures and over a number of different image datasets. For each trained network we measure the sparsity with layer granularity and the corresponding memory footprint assuming single precision float representation of each parameter. Our method is implemented in Julia language and experiments are performed using the *Knet* package [22].

### 4.1    LeNet300 and LeNet5 on MNIST

To start with, we experiment training the fully connected LeNet300 and the convolutional LeNet5 over the standard MNIST dataset [23] (60k training images and 10k test images). We use SGD with a learning parameter $\eta = 0.1$, a regularization factor $\lambda = 10^{-5}$ and a thresholding value $T = 10^{-3}$ unless otherwise specified. No other sparsity-promoting method (dropout, batch normalization) is used.

Table 1 reports the results of the experiments over the LeNet300 network in two successive moments during the training procedure.[2] The top-half of the table refers to the network trained up to the point where the error decreases to 1.6%, the best error reported in [9]. Our method achieves twice the sparsity of [9] (27.8x vs. 12.2x compression ratio) for comparable error. The bottom-half of the table refers to the network further trained up to the point where the error settles around 1.95%, the mean best error reported in [10, 16, 24]. Also in this case, our method shows almost doubled sparsity over the nearest competitor for similar error (103x vs. 68x compression ratio of [16]).

Table 2 shows the corresponding results for LeNet-5 trained until the Top-1 error reaches about 0.77% (best error reported by [9]).

In this case, when compared to the work of Han *et al.*, our method achieves far better sparsity (51.07x vs. 11.87x compression ratio) for a comparable error. We observe how in all the previous experiments the largest gains stem from the first fully connected layer, where most of the network parameters lie. However, if we compare our results to other state-of-the-art sparsifiers, we see that our technique does

Table 2: LeNet5 network trained over the MNIST dataset

| | Remaining parameters | | | | | Memory | $\frac{|\theta|}{|\theta_{\neq 0}|}$ | Top-1 |
| | Conv1 | Conv2 | FC1 | FC2 | Total | footprint | | error |
|---|---|---|---|---|---|---|---|---|
| Han *et al.* [9] | **66%** | 12% | 8% | **19%** | 36.28k | 145.12kB | 11.9x | 0.77% |
| Prop. ($S^{unspec}$) | 67.6% | **11.8%** | **0.9%** | 31.0% | **8.43k** | **33.72kB** | **51.1x** | 0.78% |
| Prop. ($S^{spec}$) | 72.6% | 12.0% | 1.7% | 37.4% | 10.28k | 41.12kB | 41.9x | 0.8% |
| Louizos *et al.* [13] | 45% | 36% | 0.4% | 5% | 6.15k | 24.6kB | 70x | 1.0% |
| SWS [10] | N/A | N/A | N/A | N/A | 2.15k | 8.6kB | 200x | 0.97% |
| Sparse VD [16] | 33% | **2%** | **0.2%** | 5% | **1.54k** | **6.16kB** | **280x** | **0.75%** |
| DNS [24] | **14%** | 3% | 0.7% | **4%** | 3.88k | 15.52kB | 111x | 0.91% |

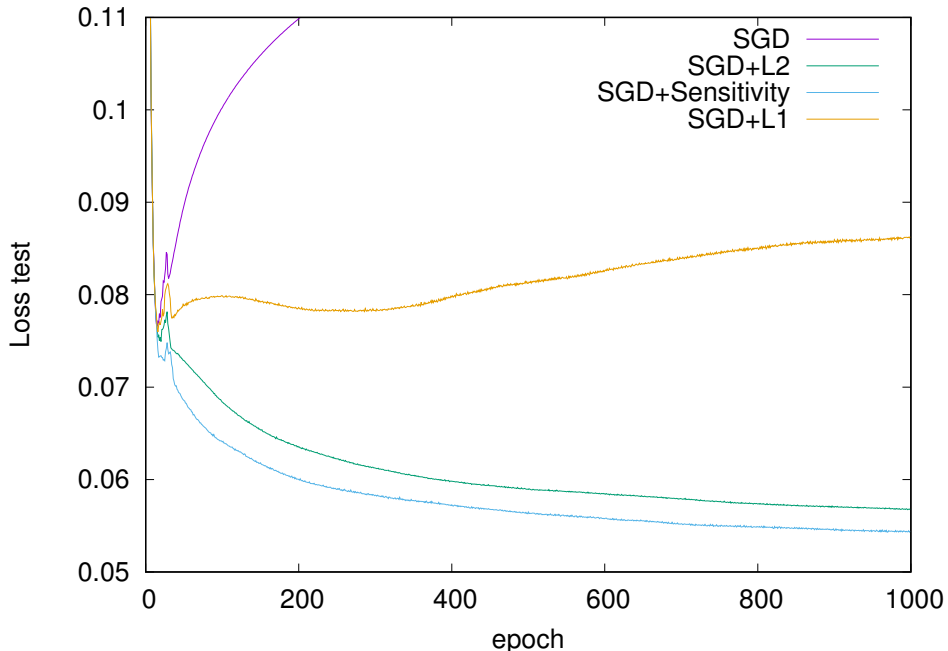

Figure 2: Loss on test set across epochs for LeNet300 trained on MNIST with different regularizers (without thresholding): our method enables improved generalization over $l_2$-regularization.

not achieve the highest compression rates. Most prominently, Sparse VD obtains higher compression at better performance compression rates. as is the case of convolutional layers.

Last, we investigate how our sensitivity-based regularization term affects the network generalization ability, which is the ultimate goal of regularization. As we focus on the effects of the regularization term, no thresholding or pruning is applied and we consider the *unspecific* sensitivity formulation in (17). We experiment over four formulations of the regularization term $R(\theta)$: no regularizer ($\lambda = 0$), weight decay (Tikhonov, $l_2$ regularization), $l_1$ regularization, and our sensitivity-based regularizer. Fig. 2 shows the value of the loss function $L$ (cross-entropy) during training. Without regularization, the loss increases after some epochs, indicating sharp overfitting. With the $l_1$-regularization, some overfitting cannot be avoided, whereas $l_2$-regularization prevents overfitting. However, our sensitivity-based regularizer is even more effective than $l_2$-regularization, achieving lower error. As seen from (16), our regularization factor can be interpreted as an *improved* $l_2$ term with an additional factor promoting sparsity proportionally to each parameter's insensitivity.

## 4.2 VGG-16 on ImageNet

Finally, we experiment on the far more complex VGG-16 [1] network over the larger ImageNet [25] dataset. VGG-16 is a 13 convolutional, 3 fully connected layers deep network having more than 100M parameters while ImageNet consists of 224x224 24-bit colour images of 1000 different types

of objects. In this case, we skip the initial training step as we used the open-source keras pretrained model [1]. For the sparsity step we have used SGD with $\eta = 10^{-3}$ and $\lambda = 10^{-5}$ for the specific sensitivity, $\lambda = 10^{-6}$ for the unspecific sensitivity.

As previous experiment revealed our method enables improved sparsification for comparable error, here we train the network up to the point where the Top-1 error is minimized. In this case our method enables an 1.08% reduction in error (9.80% vs 10.88%) for comparable sparsification, supporting the finding that our method improves a network ability to generalize as shown in Fig. 2.

Table 3: VGG16 network trained on the ImageNet dataset

| | Remaining parameters | | | Memory | $\frac{|\theta|}{|\theta_{\neq 0}|}$ | Top-1 | Top-5 |
| | Conv | FC | Total | footprint | | error | error |
|---|---|---|---|---|---|---|---|
| Han *et al.* [9] | **32.77%** | 4.61% | 10.35M | 41.4 MB | 13.33x | 31.34% | 10.88% |
| Prop. ($S^{unspec}$) | 64.73% | 2.9% | 11.34M | 45.36 MB | 12.17x | **29.29%** | **9.80%** |
| Prop. ($S^{spec}$) | 56.49% | **2.56%** | **9.77M** | **39.08 MB** | **14.12x** | 30.92% | 10.06% |

## 5 Conclusions

In this work we have proposed a sensitivity-based regularize-and-prune method for the supervised learning of sparse network topologies. Namely, we have introduced a regularization term that selectively drives towards zero parameters that are less sensitive, i.e. have little importance on the network output, and thus can be pruned without affecting the network performance. The regularization derivation is completely general and applicable to any optimization problem, plus it is efficiency-friendly, introducing a minimum computation overhead as it makes use of the Jacobian matrices computed during backpropagation.

Our proposed method enables more effective sparsification than other regularization-based methods for both the *specific* and the *unspecific* formulation of the sensitivity in fully-connected architectures. It was empirically observed that for the experiments on MNIST $S^{unspec}$ reaches higher sparsity than $S^{spec}$, while on ImageNet and on a deeper neural network (VGG16) $S^{spec}$ is able to reach the highest sparsity.

Moreover, our regularization seems to have a beneficial impact on the generalization of the network.

However, in convolutional architectures the proposed technique is surpassed by one sparsifying technique. This might be explained from the fact that our sensitivity term does not take into account shared parameters.

Future work involves an investigation into the observed improvement of generalization, a study of the trade-offs between specific and unspecific sensitivity, and the extension of the sensitivity term to the case of shared parameters.

**Acknowledgments**

The authors would like to thank the anonymous reviewers for their valuable comments and suggestions. This work was done at the Joint Open Lab Cognitive Computing and was supported by a fellowship from TIM.

## Footnotes

[1]According to our notation, $\theta = \cup_{n=1}^N \mathbf{w}_n$

[2] $\frac{|\theta|}{|\theta_{\neq 0}|}$ is the *compression ratio*, i.e. the ratio between number of parameters in the original network (cardinality of $\theta$) and number of remaining parameters after sparsification (the higher, the better).

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
