[Reviews · NeurIPS 2018]

Reviewer 1



This paper studies the sensitivity-based regularization and pruning of neural networks. Authors have introduced a new update rule based on the sensitivity of parameters and derived an overall regularization term based on this novel update rule. The main idea of this paper is indeed novel and interesting. The paper is clearly written. There are few concerns about the simulation results. 1- While the idea introduced in this paper is novel for modern deep learning, the general idea of sensitivity-based regularization for neural networks has been previously studied. As an example, authors could cite “A New Sensitivity-Based Pruning Technique for Feed-Forward Neural Networks That Improves Generalization” by Mrazova et al. published in IJCNN 2011. 2- One important question here is choosing the regularization factor and threshold value. Is there any systematic way for choosing these parameters such as cross validation? Or are these parameters tuned heuristically? 3- Regarding simulation results, it is surprising that authors have not included the result from [14] in Table 2. According to the Table 1 of [14], variational dropout (VD) outperforms [9] the pruning method of [9] significantly. Also, according to this table VD outperforms the method introduced in this paper as well. However, authors have not included VD in Table 2 and have mistakenly reported 0.8% as the best Top-1 error reported by [9] ignoring 0.75% reported by VD in [14]. The same concern holds for Table 3, as well. 4- Minor comments: -- Latex has a nice command (???). It is recommended that authors use this command instead of writing eq. 9. -- Line 235, Fig. 2 shows -- Line 239, factor can be interpreted ------------------------------------------------------------- The rebuttal did not address concerns on simulation results. In particular, authors should demonstrate how their method compares against other approaches on convolutional layers. This would make the manuscript much stronger.

Reviewer 2



Update to rebuttal: The author's should include from the Sparse VD paper in Table 2. And fix the typo that the best error was .8% when it was in fact .75% for Song's method. The author's should run on CIFAR-10 and provide results that can be compared to many other recent pruning papers. I would like to some more discussion of how many iterations the pruning procedure lasts after training. Is it comparable? 10%? The paper presents a novel approach for pruning weights for network sparsification. On two tasks (LeNet on MNIST and VGG-16 on ImageNet) it achieves higher compression than prior methods. The technique is presented as a modification of the SGD update rule, then shown to be interpretable as a regularization term, as a general form of a L2 regularizer. Overall, I find this to be a well written, easy to follow and (most importantly) useful paper. The technique would be easy to add to existing code / frameworks and seems to achieve state-of-the-art results. I think the paper should include more details on how long the sparsification phase lasts relative to the first phase, ie how much training time is added due to this procedure? Does it work just as well with optimizers other than SGD? Vanilla SGD is rarely used these days... How were the regularizing factor and threshold determined (any useful guidelines, is a grid search required)? I appreciate the derivation of (15) from the update rule, however it would also be interesting to see an attempt to explain term for other activation functions. Is there any further intuition to be had beyond the relationship to l2? My other main suggestion would be to try the technique on problems where we don't expect them to be quite so over-parameterized to begin with. That would set better expectations on what is possible for more modern architectures. It would also be insightful to see results on a RNN, as prior work has suggested they are harder to sparsify than the convnets examined in the paper (while at the same time, more likely to benefit computationally). Line 235: sows -> shows (I would also change over training to while training or during training). There are a variety of other recent sparsification techniques that are not mentioned / compared against, such as (sorry titles are not complete): "To prune or not to prune", "L0 regularization", "Fisher pruning", which makes it hard to be sure how this technique compares against all others.

Reviewer 3



The paper addresses the actual problem of structural sparsity of deep neural networks (NNs). The authors propose a general training procedure aiming for sparsity that measures relevance of a model parameter (weight) as analytical output sensitivity with respect to the particular weight. - The approach comprises a novel regularization technique that gradually lowers the absolute values of weights with low sensitivities, while weights with high sensitivities are not modified. After each training epoch, weights lower then a given threshold are zeroed out. - The authors also present a theoretical explanation and analysis of the method and its brief experimental evaluation on MNIST and ImageNet data sets. Clarity: - The presentation is comprehensible and well-organized, the new method and experiments are described adequately. - The authors also summarize some the major recent related works. However, they omit existing works concerning regularization and pruning of neural networks using sensitivity analysis (e.g., [1,2]). Typos: line 78 "rule.Then" (missing gap) line 224 "over [9] (... 68x compression ratio) over the nearest" ... 68x corresponds to the 5th row of Table 1, labeled by SparseVD[14] (not Han et al. [9]) Quality: - The paper has a fairly good technical quality. The explanation, derivation and theoretical analysis of the new method are detailed and comprehensible. However, I missed a greater experimental evaluation, especially concerning more complex NN-architectures. It is unusual, that the authors first propose the adaptation rule (Eq. 9) and after that they derive the corresponding error function (eq. 15). The general shape of the penalization function (eq. 15) is therefore quite complex and, moreover, different activation functions thus lead to different error terms. Usually, the process is opposite: to design an error function first and then the corresponding adaptation rules. Novelty and significance: The paper presents a new stochastic regularize-and-prune technique that is an interesting extension and a concurrent of weight decay. Contrary to weight decay, it doesn't reduce all of model weights but only weights with great "insensitivity", which may prevent the model from undesirable reduction of important weights. (+) The new method overcomes the reference pruning techniques on MNIST data by reaching twice the sparsity for similar error rates, while the proposed regularization technique itself proved to improve generalization ability of the NN-model and prevent overfitting (it seems that it slightly overcomes weight-decay in this respect). (-) However, the experimental results for more complex NN-architecture (VGG-16) show only relatively small benefit over the baseline technique. Therefore, more experimental results for complex tasks would be beneficial. Questions: - Based on the presented experimental results, it seems that the method prunes more extensively layers closer to the input layer. Have you observed and investigated this possible property? - A further important question is, what is the computational complexity of the algorithm (e.g., when compared to weight-decay). [1] Engelbrecht, A. and Cloete, I. A Sensitivity Analysis Algorithm for Pruning Feedforward Neural Networks. In: IEEE International Conference in Neural Networks, Washington, DC, USA. 1996. Vol. 2 of IEEE ICNN’96, pp. 1274–1277. [2] Mrazova, I., Kukacka, M. Can Deep Neural Networks Discover Meaningful Pattern Features?, Procedia Computer Science, Vol. 12, 2012, pp. 194-199. _________________________________________________________________ The rebuttal addressed most of my points (recent works, complexity) except the point that the adaptation rules seem to be the same for all activation functions. I think, the method is worth of publication, although the authors should consider more previous works and alter the paper by deeper experimental evaluation.